# Learning Disentangled Behavior Embeddings

**Changhao Shi**[*]
University of California, San Diego
cshi@ucsd.edu

**Sivan Schwartz,**[*] **Shahar Levy, Shay Achvat, Maisan Abboud, Amir Ghanayim**
Technion - Israel Institute of Technology
{sivan.s,shahar86,shay.achvat,maisanabboud,amir.gh122}@campus.technion.ac.il

**Jackie Schiller**
Technion - Israel Institute of Technology
jackie@technion.ac.il

**Gal Mishne**
University of California, San Diego
gmishne@ucsd.edu

## Abstract

To understand the relationship between behavior and neural activity, experiments in neuroscience often include an animal performing a repeated behavior such as a motor task. Recent progress in computer vision and deep learning has shown great potential in the automated analysis of behavior by leveraging large and high-quality video datasets. In this paper, we design Disentangled Behavior Embedding (DBE) to learn robust behavioral embeddings from unlabeled, multi-view, high-resolution behavioral videos across different animals and multiple sessions. We further combine DBE with a stochastic temporal model to propose Variational Disentangled Behavior Embedding (VDBE), an end-to-end approach that learns meaningful discrete behavior representations and generates interpretable behavioral videos. Our models learn consistent behavior representations by explicitly disentangling the dynamic behavioral factors (pose) from time-invariant, non-behavioral nuisance factors (context) in a deep autoencoder, and exploit the temporal structures of pose dynamics. Compared to competing approaches, DBE and VDBE enjoy superior performance on downstream tasks such as fine-grained behavioral motif generation and behavior decoding.

## 1 Introduction

Understanding the relationship between animal behavior and neural activity is a long-standing goal in neuroscience. To this end, recent advances in deep learning and computer vision has led to significant progress in the essential task of automatic analysis of high-resolution behavioral videos.

To extract behavioral information from rich video recordings, two avenues of research relying on deep learning have been proposed: landmark-based pose estimation methods [22, 26, 32] and autoencoder-based dimensionality reduction methods [2]. Pose estimation methods characterize animal behavior with the trajectories of body-part landmarks. Such methods train deep neural networks to predict manually labeled landmarks from raw video frames, then use the trained models to generate trajectories for new videos. While these methods have been highly successful in behavior analysis, they 1) require manual labeling which is often expensive; 2) heavily rely on the manual selection of the landmarks, which may differ across human annotators; and 3) struggle with subtle behavior that is hard to track, such as facial movements.

---

[*]Equal contributions

35th Conference on Neural Information Processing Systems (NeurIPS 2021).

The complementary approach seeks to reduce the high-dimensional video data to low-dimensional latent factors through non-linear dimensionality reduction. These methods [2] rely on deep autoencoders to encode video frames by solving an unsupervised reconstruction task. The learned latent embeddings can then be used for downstream tasks, such as clustering, neural decoding, etc. While unsupervised learning does not rely on manual labeling, existing approaches currently suffer from certain drawbacks. First of all, canonical autoencoders can be easily biased by the varying visual nuisances across videos, such as lighting, distance to camera and physical attributes of animals, resulting in substantial distributional shifts among the learned behavior embeddings. While non-behavioral visual variability is usually negligible within single session videos, it makes downstream analysis difficult when applied to more than one session and to different animals. Secondly, end-to-end training on videos becomes challenging when considering temporal structures, as traditional temporal modeling methods are mostly designed for low-dimensional data. How to better exploit temporal structures of high-dimensional videos for action recognition, motion planning, etc., remains an open problem for computer vision research.

In this paper, we design Disentangled Behavior Embedding (DBE) to learn robust behavior embeddings from large, unlabeled, multi-session videos. Inspired by previous works [33], DBE mitigates the distributional gaps in multi-session videos by explicitly disentangling the dynamic behavioral factors (pose) from time-invariant, non-behavioral factors (context). A video frame is thus represented by a pair of disentangled pose and context components in contrast to a single entangled representation. For a given video, the context component is designed to be time-invariant to exclude any behavioral dynamics, whereas the pose component is bottlenecked to keep it from duplicating context information. DBE latents can be used in conjunction with temporal analysis models, e.g., Variational Animal Motion Embedding (VAME) [21], to perform downstream analysis. We also propose Variational Disentangled Behavior Embedding (VDBE), a fully end-to-end trainable model, which further exploits the temporal structures of the pose components using a stochastic dynamic model. Using variational inference, a Gaussian mixture prior is trained to capture the multi-modality of the transition between consecutive pose components and generate both continuous and discrete representations of the underlying animal behaviors in videos. Our methods are not only fully unsupervised, but also enjoy superior performances on downstream tasks such as behavioral state estimation and fine-grained behavioral motif generation. To summarize, the main contributions of this paper are:

1. We develop DBE to tackle the distributional shift across multi-session videos with adaptation to the behavioral neuroscience setting, e.g., extension to multi-view videos, proper design of context embedding with standard behavioral neuroscience paradigms.

2. On top of DBE, we design VDBE to learn both continuous and discrete latent representations for simultaneous embedding and segmentation within the same model. VDBE is end-to-end trainable in an unsupervised fashion, alleviating the need to train a second post-hoc model applied to latent embeddings.

## 2 Related Works

### 2.1 Video disentanglement

Disentanglement of the pose and context in a video is an area of increasing interest in computer vision. The explicit separation of the pose from the context not only facilitates the analysis of the temporal dynamics but also makes video manipulation easier. As a pioneer in this field, MCNet [30] is an end-to-end framework that models motion (pose) and content (context) independently. MCNet uses a convolutional LSTM to model temporal dynamics and shows good performance on pixel-level video prediction on real-world datasets. Sharing similar ideas, DRNet [5] also separates the pose from the content but models temporal dynamics directly on low-dimensional pose embeddings. It also utilizes an explicit adversarial loss for disentanglement. Although DRNet shows superior performance in video prediction, it is not end-to-end trainable. DDPAE [13] tackles videos of multiple targets on the moving MNIST dataset and performs disentanglement on each individual target. Yingzhen and Mandt [33] proposed a disentangled VAE and shows controllable video generation on the synthetic Sprite dataset. MoCoGAN [29] employs a generative adversarial network that decomposes the random noises into a content part and a motion part for video generation. S3VAE [34] learns disentangled context and pose representations of videos with self-supervised regularization. In our approach we

build a disentanglement mechanism in a similar paradigm as [33], by fixing context embeddings across time and constraining the dimension of pose embeddings.

## 2.2 Stochastic dynamic modeling

Latent state-space models (SSM) are widely used to model the temporal structures for sequential data [3, 8, 17, 20, 7]. While these models have achieved fruitful results in many areas such as natural language processing and speech recognition, adapting these models to high-dimensional, non-synthetic videos is still challenging to this date. Hafner et al. [11] proposed PlaNet, with a recurrent state-space model at its core, to learn environment dynamics from high-dimensional, synthetic videos and auxiliary action variables for planning. Franceschi et al. [9] proposed a latent dynamical model enhanced by residual connections for stochastic video prediction. Although there are other papers on stochastic video generation and prediction on low-dimensional frame embeddings [1, 6, 18], these methods tackle the problems in an auto-regressive fashion and thus offer no single latent state vector that captures all the dynamics, as opposite to state-space models. Our approach closely relates to [11, 9], but differs by using a multi-modal distribution for the transition model and the ability to generate both continuous and discrete representations.

## 2.3 Behavioral video analysis

Recently, supervised deep pose estimation methods have been widely used for behavior analysis for many animal models. Methods such as DeepLabCut (DLC) [22], DeepPoseKit [10] and LEAP [26] train deep neural networks to produce confidence maps for predefined landmark annotations, all in a similar fashion. Leveraging the success of DeepLabCut, Wu et al. [32] improved the quality of pose estimation by imposing additional spatial and temporal constraints on multiple landmarks during training. Luxem et al. [21] further developed a recurrent variational autoencoder (VAME) to encode the pose estimation from DLC to a higher-level behavioral space. Prior to deep learning approaches, Kabra et al. [15] provided a supervised pipeline to automatically annotate animal behavior based on user-defined labeling.

On the other hand, unsupervised dimensionality reduction methods like PCA have been used to analyze behavior for decades [28, 4, 31]. More recently, BehaveNet [2] presented meaningful results of behavior segmentation and neural decoding. BehaveNet first uses a deep convolutional autoencoder to compress behavioral videos, then fits an autoregressive hidden Markov (ARHMM) model to exploits the temporal structures of the learned frame embeddings. However, BehaveNet was evaluated only on videos in a single experimental session and cannot handle the distributional shift across multiple sessions. Our approach, which falls into this second category, not only inherits the advantage of being unsupervised but also handles the distributional gaps across multiple sessions by explicitly disentangling poses from contexts.

# 3 Methods

## 3.1 Problem formulation

Formally, given a behavioral video $x_{1:T}$ of length $T$, we are interested in disentangling the dynamic behavioral factors from the time-invariant non-behavioral nuisances in the video in an unsupervised fashion. Our goal is to learn meaningful embedding $z^p_{1:T}$ that capture common behavioral structures across multiple animals in multiple recording sessions. We assume that the animal is performing a repetitive behavioral protocol, such as a mouse performing a forepaw reach or lever press [19, 24], and not freely moving. Additionally, in behavior experiments, videos are often recorded from different angles, resulting in multi-view data, e.g. $(x_a, x_b)_{1:T}$ where $a$ and $b$ denote the two views. For multi-view videos, we treat each view as a separate image and fuse the outputs of encoders of each view as the representation of this timestamp, whereas for mono-view videos the embedding of the frame is the natural representation. For the rest of this paper, we will slightly abuse the notations and omit the view subscripts, e.g., $x_{1:T}$ refer to either mono-view frames or multi-view frames $(x_a, x_b)_{1:T}$, and $z_{1:T}$ refer to either the embeddings of mono-view frames or the fused embeddings of multi-view frames.

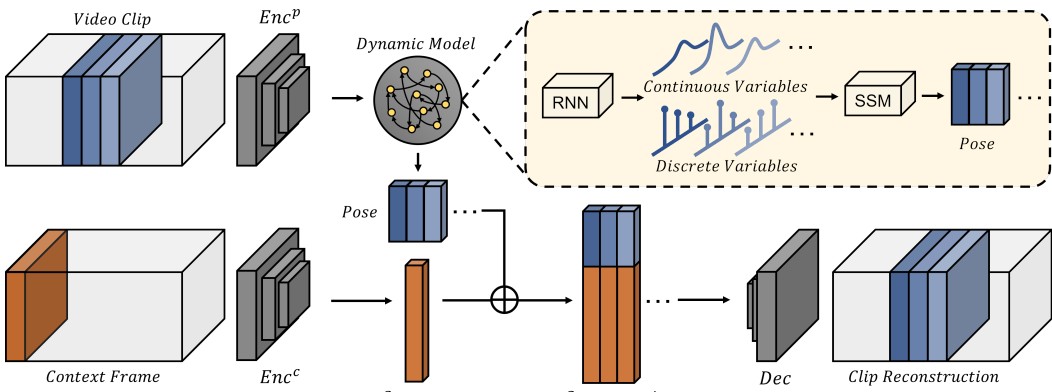

Figure 1: Overview of VDBE on a mono-view video. The pose encoder encodes individual frames into a sequence of frame embeddings, which are then passed through a stochastic dynamic model to generate pose embeddings. The context embeddings are generated from a single conditional context frame by the context encoder. Both embeddings are concatenated at each timestamp as inputs to the decoder which reconstructs the original video clip. The dynamic model (top right) uses an RNN to infer both latent continuous embeddings and discrete states that capture the transition of the frame embeddings, and uses a linear state-space model for generation. When the dynamic model is not used, the full VDBE model becomes DBE. The input video can be either single view or multi-view. For multi-view videos, encoders and decoders are used for each view separately, and the pose and context components are concatenations of the corresponding encoder outputs from each view.

## 3.2 Disentanglement

In order to factor out the non-behavioral factors, our approach utilizes two separate lines of encoders, a context encoder $E^c$ and a pose encoder $E^p$, for each view of the videos. Assuming the non-behavioral factors are time-invariant within each video, the convolutional context encoder $E^c$ infers a video-level context embedding $z^c$ from a single frame of the video. For multi-view videos, the context of each view is further concatenated into a single context embedding $z^c$, then used for the reconstruction of all the frames in this video. The pose encoder $E^p$ shares the same architecture with $E^c$, and encodes the behavioral dynamics of the animal from all video frames $x_{1:T}$ into the pose embeddings $z^p_{1:T}$. For multi-view videos, the outputs of pose encoders are concatenated to get $z^p_{1:T}$. The temporal structures of the pose component are exploited by a stochastic dynamic model stacked on top of the pose encoders, which we will detail in the following. To reconstruct the input video $x_{1:T}$, the context embedding $z^c$ is duplicated and concatenated with each pose embedding $z^p_{1:T}$, then for each view these concatenated embeddings are further decoded into frame reconstructions by the corresponding convolutional decoders $D$.

Because the context embedding $z^c$ is time-invariant and therefore incapable of capturing dynamics, we ensure that it contains no behavioral information about the animal. Thus, the key question about the disentanglement is how to ensure that the pose embeddings contain no context information. To enforce disentanglement, we create a bottleneck on the pose embeddings inspired by [33]. We choose the dimension of the pose embeddings to be as small as possible in the premise of good reconstruction. This ensures that the capacity of pose embeddings is only sufficient for capturing the temporal dynamics without any redundant context information. As we will show in the Sec 4, the disentangled pose embeddings of different sessions are well-aligned with each other, in sharp contrast against the results from canonical autoencoders where the embeddings of different sessions form distinct clusters. The disentangled context embedding, on the other hand, capture these non-behavioral nuisances and form into clusters by sessions as expected. The full disentanglement mechanism is shown in Figure 1.

## 3.3 Dynamic model

We model the temporal dynamics of the pose embeddings $z^p_{1:T}$ using a sequential variational autoencoder. For the sake of brevity, we will use $z$ to refer to $z^p$ in this section. Similar abbreviations are applied for other variables as well. We assume that the pose components govern the latent state of the behaviors and we want to model the temporal transition of the states. In a typical stochastic

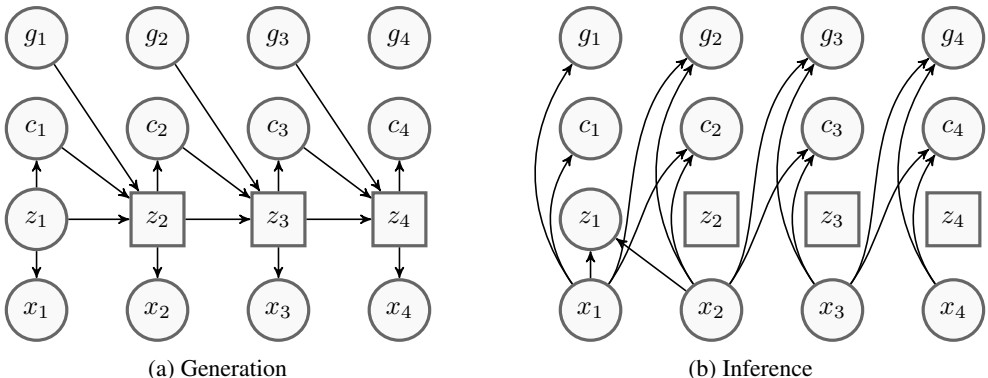

(a) Generation            (b) Inference

Figure 2: The generation (a) and inference (b) process of our stochastic dynamic model. Round nodes represent stochastic variables and squared nodes represent deterministic variables. Note that in this illustration, $z_1$ is inferred from two conditional frames $x_1$ and $x_2$; and for $c_t$ and $g_t$ only the paths from their immediate past frames are shown for the sake of clarity.

state-space model, the transition from the previous state $z_{t-1}$ to the current state $z_t$ is modeled stochastically. However, a purely stochastic transition makes it hard for the model to keep long-term information [11]. Thus, we factor out the stochasticity of the dynamics into a separate component $s$. This means that at a given timestamp, the stochastic latent variable $s_t$ is first sampled from a prior distribution, usually a Gaussian distribution, then the current state $z_t$ is generated from the previous state $z_{t-1}$ and the stochasticity $s$ deterministically.

While this strategy has been proved to be effective in many scenarios [6, 11, 9], the uni-modal Gaussian transition distribution is limited for behavior analysis because it does not capture the multi-modality of behavior. For example, in the video dataset we will show in Section 4, the transition of the animal poses is usually caused by a set of distinct, prototypical actions, such as lift, reach and grab. A uni-modal Gaussian transition distribution is inadequate for modeling such multi-modal transitions. Furthermore, for the purpose of motif segmentation it offers no discrete states which are highly desired.

A more suitable choice for capturing such multi-modality is a Gaussian mixture model [27]. Shu et al. [27] design the transition distribution in the generative model to be a Gaussian mixture while keeping the posterior inference approximation to be a uni-modal Gaussian. A closed-form approximation of the exact KL divergence is then used in evidence lower bound for optimization. In this paper, we use an alternative approach to introduce the Gaussian mixture. Instead of using a single stochastic latent variable $s$ with a Gaussian mixture prior, we decompose it into two stochastic variables, the mixture index $c$ and a Gaussian noise $g$. This Gaussian mixture prior also enables the model to perform embedding ($z_t$) and segmentation ($c_t$) simultaneously. With all these, we can dive into the details of our variational autoencoder, as illustrated in the zoomed-in region in Figure 1.

**Generation** The generative procedure is shown in Figure 2a. We model the transition between the previous state $z_{t-1}$ and the current state $z_t$ using a Gaussian mixture. The generative model is:

$$z_1 \sim \mathcal{N}(0, I), \ z_{t+1} = f_\theta(z_t, s_t), \tag{1}$$
$$c_t \sim \mathrm{cat}(\pi_\theta(z_t)), \ g_t \sim \mathcal{N}(0, I), \ s_t = \mu_{c_t} + \sigma_{c_t} \odot g_t, \tag{2}$$
$$x_t = \mathcal{N}(\mathrm{Dec}_\theta(z_t, z_c)), \tag{3}$$

where $\pi_\theta$ and $f_\theta$ are linear projection layers. At each timestamp, stochastic variables $c_t$ and $g_t$ model the discrete state and the additional stochasticity, then the continuous state $z_t$ is generated with a deterministic path from $z_{t-1}$, $c_t$ and $g_t$. Note that although both $c_t$ and $g_t$ capture the stochasticity of the transition, only $c_t$ is dependent on the previous state $z_{t-1}$. $c_t$ is sampled from a categorical distribution parameterized by $z_{t-1}$ and $g_t$ is simply sampled from a standard Gaussian distribution. The inferred $c_t$ is used for for motif segmentation in the test stage. We assume the covariance matrix of each Gaussian component is diagonal and $\odot$ denotes element-wise multiplication (Hadamard product).

**Inference** Because of the non-linearity in the generative model, the true posterior distributions of the latent variables are intractable. As shown in Figure 2b, we use variational inference to approximate these posteriors. The inferred posterior is factorized as $q_\phi(c_{1:T}, g_{1:T}, z_1) = \prod_{t=1}^{T} q(c_t|x_{1:T})q(g_t|x_{1:T})q(z_1|x_{1:C})$, where $q(c_t|x_{1:T})$ is a conditional categorical distribution, and $q(g_t|x_{1:T})$ as well as $q(z_1|x_{1:C})$ is a conditional Gaussian distribution. We use a shared GRU network followed by separate projection heads for $q(c_t|x_{1:T})$ and $q(g_t|x_{1:T})$. For $q(z_1|x_{1:C})$, we use an MLP whose inputs are the first $C$ conditional frames in the sequence. At test-time, we also use these inference networks to generate behavior representation, i.e. we use the mean of the Gaussian distribution and the highest category of the categorical distribution as the continuous and discrete representations of the behavior of each frame.

**Optimization** We use the following evidence lower bound (ELBO) for optimization:

$$
\mathcal{L}_{\theta,\phi}(x_{1:T}) = \sum_{t=1}^{T} \left[ \mathbb{E}_{q_\phi(c_{1:T}, g_{1:T}, z_1)} \log p_\theta(x_t|z_t) - \alpha \mathcal{D}_{KL}(q_\phi(c_t|x_{1:t})||p_\theta(c_t|z_{t-1})) \right.
$$
$$
\left. - \beta \mathcal{D}_{KL}(q_\phi(g_t|x_{1:t})||p_\theta(g_t)) \right] - \gamma \mathcal{D}_{KL}(q_\phi(z_1|x_{1:C})||p_\theta(z_1)),
$$
(4)

where $\alpha$, $\beta$ and $\gamma$ are the trade-off parameters that control the information flow to each stochastic latent variables $c_{1:T}$, $g_{1:T}$ and $z_1$ respectively [12]. The first log-likelihood term corresponds to the frame reconstruction and the other KL-divergence terms are the regularization of the inferred stochastic latent variables to their prior knowledge.

The above expectation is estimated with a single sample from the approximated posterior distribution. To allow gradients to back-propagate through the non-differentiable sampling operation, we use the re-parameterization tricks. For the Gaussian posterior distributions $q(g_t|x_{1:T})$ and $q(z_1|x_{1:C})$, we re-parameterize the distribution into their estimated mean and standard deviation with a standard Gaussian noise [16]. For the categorical posterior distribution $q(c_t|x_{1:T})$ we use the Gumbel-Softmax re-parameterization trick [14]: the sampling process is re-parameterized into the estimated discrete log-probability and a Gumbel noise, following by a softmax normalization. The outputs of softmax are also discretized to one-hot encodings to ensure consistency between training and testing. Note that theoretically one can compute the exact expectation over the discrete latent variable $c_t$ without using Monte Carlo estimation, but our experiments show that one-time sampling can generate satisfying results with the advantage of being more efficient.

## 4 Results

We evaluate our methods on two multi-session video datasets: the Hand-reach dataset [19] we collected, and the Wide-Field Calcium Imaging (WFCI) dataset [24, 25]. The Hand-reach dataset contains videos of head-fixed mice who were trained to reach from their perch to a food pellet. The datasets consists of 263 successful trials of 3 expert animals (who have learned the motor task) in 8 sessions, and includes fine-grained behavior. In a successful execution of the task, the animal will lift its paw from the perch, reach and grab the pellet and bring it to its mouth and eat it. The animal might perform multiple grab attempts in a single trial before successfully grabbing the pellet. An illustration of the Hand-reach dataset is shown in Figure 3. The videos were acquired on multiple dates, within which the experimental layout is potentially different, e.g., location of the table. Two cameras simultaneously record from the front and right side of the mouse. Each video is 12 seconds long, with a frame rate of 200Hz. The video

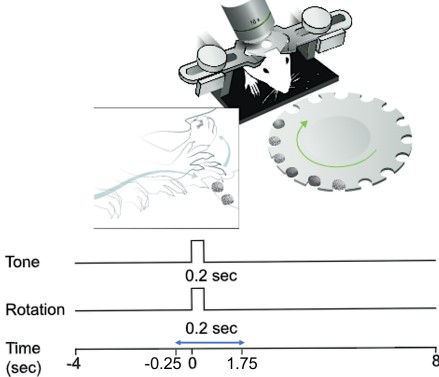

Figure 3: Hand-reach dataset. Mice were head-fixed and trained to grab a food pellet. Trial structure: 4 seconds ITI period followed by the tone and table turning. We analyze the [-0.25, 1.75] segment which contains most of the significant behaviors for expert mice.

begins with a 4-seconds inter-trial interval (ITI), the end of which we set as the origin (0) of our time axis. A cue of 0.2 seconds is shortly followed by table-turning which brings the food in front of

the mouse. Since the mice in this dataset are experts, and thus mostly stationary before the cue and chewing the pellet after successful execution of the hand-reach, we use a 2-second clip of the main dynamic behavior of each full video to train the model. Each clip starts at 0.25 second before the cue and ends at 1.75 seconds after, which contains most of the motions of interest. The context frame for each video is selected from the ITI period, sampled at 1 second before the cue, when most of the subjects are stationary.

In the WFCI dataset (an extended version of the dataset in [2]), head-fixed mice perform a visual decision making task, using levers to initiate trials and reporting choices by licking from spouts. The behavioral videos are recorded from a side and a bottom view of the subjects, with a length of 189 frames and a frame rate of 30Hz. The dataset contains 3708 videos from 8 sessions of 4 animals in total. We randomly select $80\%$ of the videos as training set and use the remaining $20\%$ as test set.

We train both DBE and VDBE, without and with the stochastic dynamic modeling component. We compare our methods against two behavior embedding methods: BehaveNet [2] and VAME [21], with implementation details of each provided in Appendix B and Appendix C, respectively. In Section 4.1 and 4.2, we demonstrate the efficacy of our disentanglement component. In Section 4.3, we show both DBE (combined with a motif segmentation model) and VDBE achieve superior results for motif segmentation compared to the original models.

## 4.1   Disentanglement

To demonstrate the efficacy of disentanglement, we focus on session recognition: given the behavior embedding of a specific video frame, can we identify which session (animal and date) this frame comes from? We train a simple linear classifier to classify the experimental session from different pose embeddings, and ideal disentanglement should generate session-agnostic pose embeddings. The accuracy of the session classification for each model is reported in Table 1. DBE as well as our full VDBE model generates session-agnostic pose embeddings, compared to the plain CAE autoencoder used in Behavenet. Although DLC also provides session-agnostic coordinates, selected markers do not encode enough useful information for extracting robust behavioral motifs across sessions (Section 4.3). The context embedding of our methods (DBE-context and VDBE-context) accurately classify the session, further demonstrating disentanglement of context from pose.

We further visualize the advantage of disentanglement in Figure 4. We plot the 2D PCA applied to the BehaveNet and DBE embeddings of two videos that show similar behaviors but are recorded in different sessions.The plain CAE of BehaveNet generates separate pose embeddings even though the behaviors are similar. This distributional shift will be problematic when one wants to transfer the knowledge learned in one session to another. This means that one needs to learn a new model every time when there are new sessions, and learn a mapping across animals to relate the embeddings of the same behavior. In contrast to BehaveNet, our DBE nicely aligns the two videos in the pose embedding space. As we will show in the later section, this consistency is crucial to motif segmentation and behavior decoding. More results comparing BehaveNet and DBE embeddings, as well as ethograms of both the Hand-reach and the WFCI dataset, are provided in Appendix F and G.

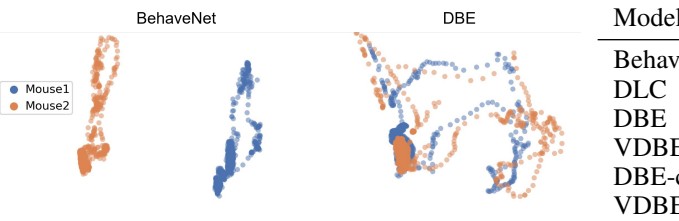

| Model | Hand-reach | WFCI |
|---|---|---|
| BehaveNet | 100.00 | 99.99 |
| DLC | 21.19 | - |
| DBE | 28.21 | 35.49 |
| VDBE | 17.94 | 19.54 |
| DBE-context | 88.59 | 100.00 |
| VDBE-context | 96.19 | 100.00 |

Figure 4: Embeddings of 2 similar videos (Hand-reach)          Table 1: Session classification

## 4.2   Trajectory decoding

DLC is a robust method used to track manually selected landmarks in behavioral videos. A good behavioral embedding should contain the information encoded by DLC. To illustrate that our approach learns meaningful behavior embeddings, we perform a simple linear regression task from the DBE

embeddings to the DLC coordinates in the Hand-reach dataset. A good fit of the linear regression model suggests that our learned embedding contains the DLC coordinate information.

We place 5 markers on the center and 4 digits of the right paw of the mouse in each view of the videos, and train a DLC model to track the coordinates of those 10 markers. Due to the differences in spatial layout of the sessions (e.g., table location), we fit separate linear regression models to each marker for each individual experimental session. Only DLC coordinates that have a confidence score larger than 0.95 are used for the training of linear regression.

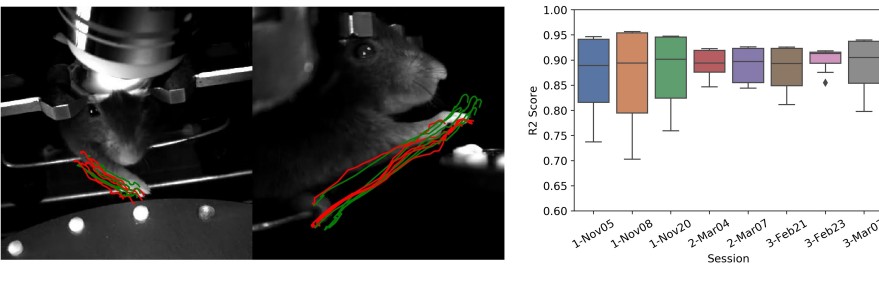

(a) DLC trajectories prediction             (b) Goodness-of-fit of linear regression

Figure 5: Predicting DLC trajectories from disentangled embeddings on the Hand-reach dataset. (a) Example video overlaid with true DLC trajectories (red) and the linear regression model predictions (green). (b) $R^2$ averaged across each coordinate for each individual session.

An example of DLC predictions and the average $R^2$ of each individual linear regression model are shown in Figure 5. In Figure 5a, the predictions are well aligned with the true DLC trajectories, capturing the right paw of the mouse moving from the perch to the pellet on the table. The high $R^2$ scores in Figure 5b show that the linear regression models are well-fitted to the data, which indicates that the DBE embeddings manifest meaningful information about the selected markers.

## 4.3   Motif segmentation

Learning fine-grained behavior motifs from videos is an essential but challenging task for analyzing animal behavior, especially when no supervision is available. In this section, we compare our approach with other competing methods for motif segmentation on the Hand-reach dataset. To evaluate the quality of the generated motifs, we select 10% of the videos (25 videos) and perform expert manual labeling of the selected videos with a set of seven behaviors of interest: On-Perch, Lift, Reach, Grab, To-mouth, At-mouth, and Back-To-Perch (4730 labeled frames). The rest of the videos are used as the training set for motif segmentation methods, i.e. clustering. For each method, we assign the learned motifs to the labeled set of behavior, where the mapping is selected to maximize the clustering accuracy of holdout videos. Thus, each motif corresponds to a "mega-cluster" that may or may not contain more than one inferred discrete state. Note that we set the number of the discrete states $k$ to be larger than the number of target labels for two reasons. The first reason is that only part of the frames are labeled and the target labels do not cover all the possible behavior of the mice. Extra clusters are necessary to capture these additional behavior. The second reason is that setting a larger number of motifs rather than the exact number of labels accounts for inter-class versus intra-class variability. For example, this enables separating a highly variable behavioral label like Reach into more than one motif without having to compensate for that by merging At-mouth and To-mouth into a single motif. We evaluate the performance of motif segmentation on three clustering metrics, accuracy, Normalized Mutual Information (NMI), and Adjusted Rand Index (ARI).

We use the DBE embeddings in two variations for motif segmentation: DBE+ARHMM which replaces the CAE embeddings of the BehaveNet model with DBE embeddings, and DBE+VAME, which replaces the DLC trajectories with DBE embeddings as inputs to VAME. As shown in Table 2 and Figure 6, disentanglement is the key to achieve good behavioral clusters across different animals and multiple sessions. Combining DBE for frame embedding and 30-cluster VAME (DBE+VAME) for temporal dynamics and segmentation achieves the best accuracy, NMI and ARI. Our full VDBE model has very similar performance, with both models exhibiting $\sim 30\%$ higher accuracy than BehaveNet and over $20\%$ higher accuracy than the original VAME framework (relying on DLC inputs). These results also show that using a more powerful temporal model such as VAME can

Table 2: Multi-session motif segmentation for the Hand-reach dataset

| Model | Unsup. | Dis. | End2End | Accuracy | NMI | ARI |
|---|---|---|---|---|---|---|
| BehaveNet (k=15) | ✓ | ✗ | ✗ | 50.52 | 30.89 | 20.98 |
| BehaveNet (k=30) | ✓ | ✗ | ✗ | 53.09 | 33.44 | 25.09 |
| DBE + ARHMM (k=15) | ✓ | ✓ | ✗ | 64.04 | 42.06 | 36.08 |
| DBE + ARHMM (k=30) | ✓ | ✓ | ✗ | 71.49 | 48.96 | 45.76 |
| DLC + VAME (k=30) | ✗ | ✗ | ✗ | 57.17 | 42.07 | 44.09 |
| DBE + VAME (k=15) | ✓ | ✓ | ✗ | 71.31 | 59.47 | 51.46 |
| DBE + VAME (k=30) | ✓ | ✓ | ✗ | 81.25 | 67.31 | 67.33 |
| VDBE (k=30) | ✓ | ✓ | ✓ | 79.53 | 64.85 | 66.94 |

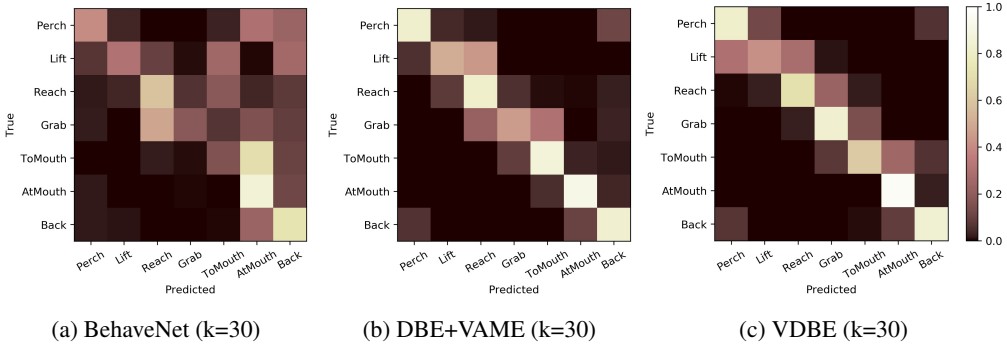

(a) BehaveNet (k=30)  (b) DBE+VAME (k=30)  (c) VDBE (k=30)

Figure 6: Confusion matrices of BehaveNet, DBE+VAME and VDBE for the Hand-reach dataset.

benefit motif segmentation. With disentanglement, the confusion matrices ( Fig. 6) are more diagonal which indicates better clustering. Figure 7 shows examples of random segments from VDBE motifs corresponding to 4 different labels. Note that VDBE successfully clusters the behaviors even when the context is varying. Additional results can be found in supplementary materials.

### 4.4 Controllable behavior generation

One advantage of VDBE and ARHMM-based models compared to VAME is that we can decode synthetic videos from behavior embeddings, which demonstrates better interpretability and allows potential applications to neural decoding. In this section, we further investigate behavior decoding by manipulating the context components. In order to generate synthetic videos that replicate source behavior with a new context, we use the inferred discrete behavior representations $c_{1:T}$ from the source video and the context embedding from the target session, while $g_{1:T}$ and $z_1$ are sampled from their prior distributions. If the target session is the same as the source session, the synthetic video should be similar to the source video. The results are shown in Figure 8. Note that the behavior in the source video is replicated in both generated videos with the same and different contexts. This indicates that our model successfully learns the dynamics of the behavior.

## 5 Conclusions

In this paper, we tackle the problem of unsupervised behavior representation learning from multi-session videos. A disentanglement mechanism is designed to separate session-wise non-behavioral factors from the useful behavioral information, and a stochastic dynamic model is jointly trained to model the temporal dynamics of behaviors. Compared with competing methods, our method shows superior performance on multiple tasks such as fine-grained motif segmentation, meanwhile also enjoys the advantages of being unsupervised, end-to-end trainable, and more interpretable.

Despite the above improvements, a couple of open questions still remain and will be explored in future work. Firstly, the current model which relies on a predefined number of states is limited in generalization of the motif assignment to videos which include novel unseen behavior. Secondly, our disentanglement assumes the non-behavioral nuisances are stationary through a video, which is not always true. Future work should benefit from allowing slow dynamics in context embeddings. Thirdly,

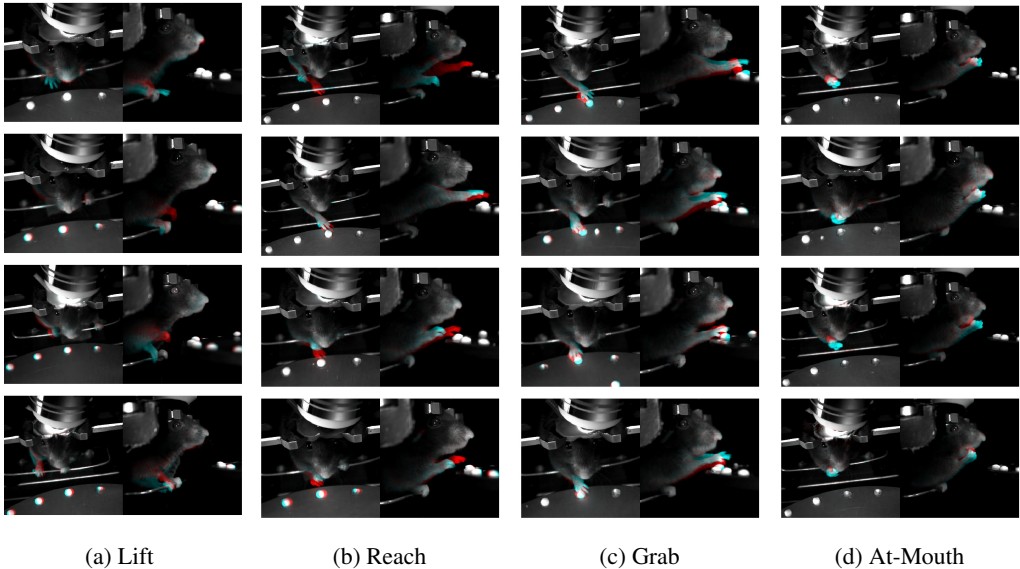

(a) Lift        (b) Reach        (c) Grab        (d) At-Mouth

Figure 7: Motifs learned by VDBE. The blue channel indicates the start of the motif and the red channel indicates the end. Note that in each column the behavior is consistent but the context is varying (e.g., location of the table and identity of the mouse).

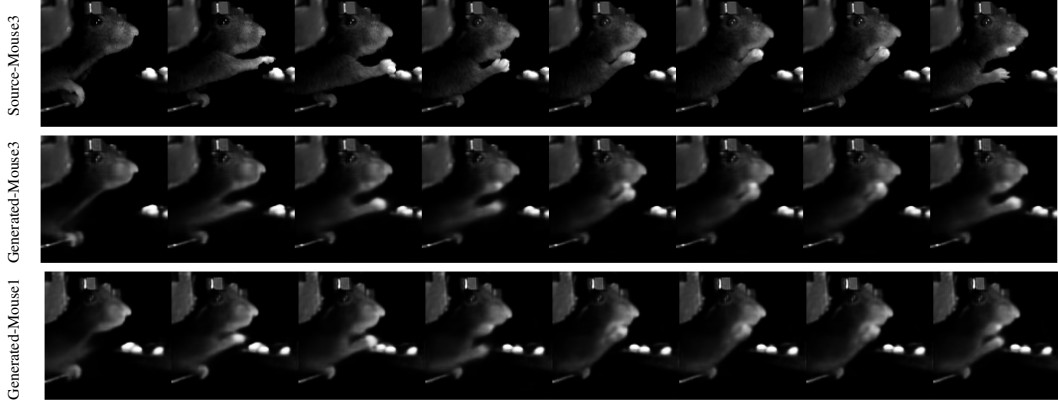

Figure 8: Source (top row) and generated videos of the same discrete behavior sequence, with the same (middle row) and different (bottom row) contexts. Each row is plotted at intervals of 20 frames.

with the distributional shift in videos confirmed, it would be interesting to study disentangling other visual behavior representations such as DLC for multi-session analysis. Finally, with the learned behavioral embeddings consistent across sessions, future works can leverage this representation to relate behavior to neural activity [23].

Regarding societal impacts of our study as well as behavior analysis research in general, although there is a significant gap between the head-fixed animal behavior studied here and naturalistic human behavior, we should always be careful about the negative consequences for human monitoring.

Code is available at https://github.com/Mishne-Lab/Disentangled-Behavior-Embedding.

## Acknowledgement

This work was supported by in part by the National Institutes of Health grant No. R01EB026936. We thank Simon Musall and Matthew Whiteway for providing the WFCI dataset and their generous help.

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
