# OpenReview forum: "Learning Disentangled Behavior Embeddings"
_NeurIPS.cc/2021/Conference — NeurIPS 2021 Spotlight_

### Official Review · Reviewer_88jd · 2021-07-13

**Rating:** 8
**Confidence:** 4

**Summary:**

The authors present an end-to-end trainable framework for the unsupervised analysis of animal behavioral videos. Their model produces a latent representation for each frame which separates pose information from context (background) information, as well as models the temporal evolution of the pose information. This model can be applied to multi-view videos with varying background information.

**Limitations And Societal Impact:**

This model was thoroughly tested on two head-fixed mouse datasets, but it is not clear how well the model will work when analyzing videos of freely moving animals. I don't think the authors need to analyze a new dataset, but they should mention this limitation (of their analyses, not necessarily the model) in the discussion.

**Main Review:**

originality: This paper combines ideas from several previous works in a seemingly novel way. It would be useful in the Related Works section if the authors could comment on how their approach differs from DRNet (DBE does not use an adversarial loss for disentanglement; are dynamics models similar?), DDPAE (not clear what similarities/differences are here), and Yingzhen and Mandt (do they model temporal dynamics?). The paper provides clear advances over the references cited in the behavioral video analysis section.

quality: The method is sufficiently tested and rigorously compared to relevant baselines.

clarity: The paper is very clearly written. Table 2 could benefit from a caption if space allows.

significance: The authors have adapted several more general video analysis models to address a specific (but important) video analysis task: the modeling of animal behavioral videos from neuroscience experiments. The results demonstrate this method clearly outperforms relevant baselines from the neuroscience literature (DLC, Behavenet, VAME) but does not compare to the more general models (DRNet, DDPAE, etc).

**Time Spent Reviewing:**

1

---

> ### Author Response · Authors · 2021-08-10
> **Response to Reviewer 88jd**
>
> We thank the reviewer for your very positive comments!
>
> Comparison to related work: Your understanding of previous methods is correct and they are all different from our method. DRNet uses an adversarial loss; DDPAE focuses on the compositionality of multiple objects; Yingzhen and Mandt (so are the other two methods) model temporal dynamics but they do not learn discrete latent representations for segmentation. Additionally, all these methods do not consider multi-view datasets. We agree that adding these differences between previous works and our methods is important and we plan to do that in the camera-ready version.
>
> Table 2: we will add a more detailed caption if space permits.
>
> Head-fixed animals: We also agree with the reviewer regarding the limitation of the model being designed for head-fixed animals, and we intend to explore freely behaving animals in future work. We will add this to the conclusions in the camera-read version as well.

---

### Official Review · Reviewer_Yzq9 · 2021-07-15

**Rating:** 7
**Confidence:** 4

**Summary:**

In this work, the authors propose a new model for analyzing “unlabeled, multi-view, high-resolution behavioral videos across different animals and multiple sessions”. This model (Disentangled Autoencoder, DisAE) learns robust behavior embeddings that are disentangled from the constant context in the experiments. The authors further add a stochastic dynamic model to DisAE to get an end-to-end approach (DBE) that both learns meaning representations and generates videos. After comparing the proposed models to previously proposed ones, the authors show that DisAE and DBE outperform earlier models on encoding less information about the experiment session and also on motif segmentation tasks. However, this paper lacks ablation studies supporting the specific designs in the models. Overall, the new models proposed in this work seem to be stronger than others and are therefore worth being known to the community.

**Limitations And Societal Impact:**

Yes, both of these are briefly addressed in the discussion section.

**Main Review:**

This work proposes a neural network model for behavior analysis on videos recorded from behavioral experiments from different animals and multiple sessions. The model, called DisAE, is designed to produce embeddings only encoding animal behaviors but not the experiment context. This is done through having one embedding that varies across different frames to encode the animal pose information and another embedding that is only determined by one frame to encode the session-specific context information. Both embeddings are then concatenated to get the final embedding that is used to reconstruct the input frames. Based on this DisAE model, the authors further propose a stochastic dynamic model to learn and infer how the pose embeddings evolve through time. This model is then added to DisAE, yielding the DBE model, which is trained end-to-end to learn behavior representations and generate new videos. To illustrate the power of these two proposed models, the authors first compare them with BehaveNet and DLC on how much the session-specific information can be decoded from the animal pose embeddings and show that DisAE and DBE indeed have less information about sessions in their pose-specific embeddings, just as designed. These two models are then compared to other models on how well they can do motif segmentation, which also yields better performance. The authors also show that DBE successfully reconstructs reasonable videos even for a different context.
Although the better performance of DisAE and DBE compared to other models is indeed significant, the paper can greatly benefit from more ablation studies to justify its design. For example, the authors have mentioned that the pose embeddings are designed to be smaller dimensional to make them only contain the dynamic information. But there are no empirical results supporting this design. The authors could have compared the performance of models with different dimensions on the relevant task (like the session information prediction task) to show this. Another design is the choice of using Gaussian mixture model instead of single Gaussian distribution in the stochastic dynamic model. Though it is theoretically reasonable to imagine how different actions could require different Gaussian distributions, an empirical experiment to support this hypothesis can still be useful.
In general, the new models proposed in this work are indeed models with better performance and therefore worth being known by the community. The paper itself is also well-written and easy to read.

**Time Spent Reviewing:**

4.5

---

> ### Author Response · Authors · 2021-08-10
> **Response to Reviewer Yzq9**
>
> Thank you for your positive comments! Your understanding is correct. It is true that the dimension bottleneck is important for disentanglement, and the previous Yingzhen & Mandt [31] paper as well as our observations show that a smaller dimension is beneficial. It is also true that the Gaussian mixture model has natural advantages over single Gaussian distribution for downstream behavior segmentation. This is essential as the motifs are determined from the cluster ids $c$ in our current model. We will clarify this in the paper. We are very interested in exploring these ablations in the future.

---

> > ### Comment · Reviewer_Yzq9 · 2021-08-30
> > **Response**
> >
> > Thanks for the response. I still recommend for acceptance and look forward to the future work!

---

### Official Review · Reviewer_bsYQ · 2021-07-16

**Rating:** 7
**Confidence:** 3

**Summary:**

Animal behavior can be a huge space full of behavioral motifs. Hand-labeling allows for algorithms to much better extract behavioral information, but it takes a lot of time. This paper tries to address this problem by building an algorithm that extracts this information from unlabeled behavioral videos. The authors evaluate the ability of the algorithm to generalize across sessions, as well as the quality of the various behavioral motifs identified by DisAE.


**Limitations And Societal Impact:**

In section 5 the authors discuss two important limitations of their work, the ability of the algorithm to generalize on unobserved behavioral motifs and the sensitivity to the stationarity of the non-behavioral factors. I would include as a limitation the fact that even if unsupervised, this algorithm needs supervision for the identification and further classification of various behavioral motifs.

**Main Review:**

In this paper the authors develop an unsupervised algorithm to extract behavioral motifs from either a hand reach task that they collected or a decision-making task from an already published work. The authors are properly motivating the question and cite relevant work adequately. They present their results with clarity and the paper is well organized.

The proposed approach (DisAE algorithm) shows great generalization performance across sessions when compared to BehaveNet. Moreover, it shows quite good performance in predicting forelimb trajectories from disentangled embeddings showing that the DisAE embeddings can carry information about various behavioral markers. Finally, when coupled with either ARHMM or VAME (or just the DBE) it has great ability to capture various behavioral motifs.

It would be nice to evaluate how the algorithm handles the time-variant non-behavioral factors such as moving bedding, visual stimulus reflections etc. In some of the example videos it can be seen that the DBE captures the movement of the food pellet disk which I expect would influence the ability of the algorithm to separate the motifs. Also, the context identification is automatic but how does it ensure that the context embedding does not contain any behavioral information? While in the text (4.4 section) it is claimed that the model successfully learns the dynamics of the behaviors based on the ability to generate the behavioral output, the included videos in the supplementary data show relatively poor performance of the DBE algorithm in generating a behavior. It would be important to provide a proper evaluation of the DBE algorithm performance by showing the various motifs that can be generated.

Overall, I find this paper good and while there are several algorithms that can be used to dissect behavioral motifs, I find that the DisAE algorithm shows great performance and once combined with other temporal analysis models it can be useful for analysis of behavioral motifs in unlabeled videos in an unsupervised manner.


**Time Spent Reviewing:**

5

---

> ### Author Response · Authors · 2021-08-10
> **Response to Reviewer bsYQ**
>
> We thank the reviewer for your positive comments!
>
> We agree that exploring time-varying non-behavioral factors is an interesting direction. Currently, we extract the context from a single frame and use this same embedding to reconstruct any frame in the video. This design enforces the context embedding to be time-invariant to exclude any dynamic behavioral information, but it would be interesting to achieve disentanglement through other approaches such as adversarial training to allow the context embedding to be time-varying.
>
> Regarding the concerns about the use of supervision after the clusters have been outputted, our method, same as other unsupervised clustering methods, indeed requires manual intervention to now “name” or label those clusters. The benefit in this setting is that our algorithm provides a meaningful segmentation that a user can quickly label as opposed to having to label frame by frame from scratch, thus greatly reducing the time burden and involvement of human annotators. In experiments where the sequence of expected executed behaviors is known, labeling the motifs post-hoc can be facilitated by using the transition matrix of motifs (see Figure 14 in the appendix).

---

### Official Review · Reviewer_d5ns · 2021-07-17

**Rating:** 7
**Confidence:** 3

**Summary:**

This paper introduces a method named Disentangled Autoencoder (DisAE) to learn disentangled pose and context representations from videos. By keeping the pose component static based on a key frame and by extracting temporal context components, the proposed DBE model starts to generate both continuous and discrete representations from the input video. Temporal dynamics of the pose components were learned using sequential variational autoencoder under a stochastic dynamic model which utilizes a gaussian mixture prior considering two stochasitic variables for the mixture index and a gaussian noise. The method was evaluated on controlled multi-session video datasets, one which the authors collected (Hand-reach) and another one is WFCI. Associated code and sample videos were also shared. However, Hand-reach dataset was not openly available.

**Limitations And Societal Impact:**

Have the authors adequately addressed the limitations and potential negative societal impact of their work? Yes

**Main Review:**

The paper highlights an interesting and important work in learning disentangled representations of animal behavior. The collected Hand-reach dataset is much appreciated; however, the dataset is not open-sourced as far as I can tell. Some comments and concerns are as below:

1.	The manuscript is fairly well written. However, the use of abbreviations and mathematical terms without proper explanations affect the readability. For example, in line 50, VAME is not defined; in line 99, ARHMM is not defined; in line 108, Z^p_(1:T) is not defined.

2.	Figure 1 is not referred in the manuscript.

3.	Figure 1 omitted the multi-view nature of the input dataset. This brings confusion on how the context and pose embeddings are generated from multiple views. There are notable differences between Section 3.2 Disentanglement and Figure 1, which makes it hard for the readers to connect the overview of DBE (Figure 1) and the methodology.

4.	In the code shared, models/nets.py presents the class definitions for BehaveNet, DisAE, and DBE. IN DisAE and DBE network class definitions, authors define a self.first_frame variable (default = True) taking input from config file. While going through the training code, there is no apparent updates provided to this variable during the optimization. If that is the case, then the if condition in lines 161, 255, and 326 will always be taken. Was this deliberate or an error in the code? From the code it looks like the embeddings are getting fused here. Am I missing something?

5.	In Figure 5, the inter quartile range associated with the R^2 score for first few sessions are considerably larger than the later sessions. Does this have any relations to how the DisAE model was trained and the session classification performance?

6.	I would like to hear the authors thoughts on how the proposed disentanglement methodology might work on large behavior datasets such as CAlMS21 (https://data.caltech.edu/records/1991) which was recently published. Multi-session multi-subject data of large scale could affect the performance of the proposed stochastic dynamic models while generating long videos. I am curious to see how the length of input videos play a part in the video generation over multiple states.

7.	It would be better if the data descriptions and hardware and software experimental setup are presented as subsections instead of as paragraphs under Section 4 - Results.

8.	It would also be better if the main hyperparameters chosen for the training are included in the main manuscript rather than in the appendix.

9.	Some information about the motif transitions could have been included in Figure 8 to compare based on the source and generated videos.



**Time Spent Reviewing:**

7 hrs

---

> ### Author Response · Authors · 2021-08-10
> **Response to Reviewer d5ns**
>
> Thank you for your helpful feedback on our paper! We would like to address your concerns correspondingly:
>
> 1. Readability: Thank you for pointing these out! We will properly define notations and acronyms in the camera-ready version.
>
> 2. Reference to Figure 1 in the text: We will fix this in the camera-ready version.
>
> 3. Multi-view data in Figure 1: Sorry for the confusion. We would like to stress that our methods can be used with mono-view data as well and the multi-view nature of the data is not the focus of our paper. When using multiple views, the pose embedding of a given timestamp is the concatenation of the outputs of 2 separate convolutional encoders which take the frame of each view respectively. Same for the context embeddings. When there is only 1 view (not exploited in the paper), the embedding is simply the output of the encoder that takes the single input. We will rephrase 3.2 and clarify this in the Figure 1 caption to make this clearer.
>
> 4. Code: You are right about the code. Our method selects a single timestamp and fuses the multi-view data from this timestamp to get the context embedding. However, note that it is not always the first frame of the video being selected but rather that a random frame from a selected period. We are sorry about this typo update delay and will correct it in the future released code.
>
> 5. Range of $R^2$ in Figure 5: We don’t find a clear relationship between the range of $R^2$ and the training. The model is trained such that we treat each session equally, and the model is trained on all sessions at once. It’s more likely due to the variability in the behaviors of the corresponding animals (note the first few sessions belong to one animal and later sessions belong to the 2 others). The linear regression model is trained separately on each session.
>
> 6. CalMS21 dataset: Thank you for pointing the new CalMS21 dataset to us! Indeed we think this is a very valuable addition to the community of computational behavioral neuroscience, but the fact that CalMS21 only contains trajectory data and that the animals behave freely in CalMS21 makes it very different from the two datasets we consider in the paper. Also, the paper accompanying this dataset provides very few video frames demonstrating the animal behavior and potential multi-session variability, so it is difficult for us to hypothesize how our approach would extend to the video data if it were made available. However, we intend in future work to extend our approach to freely behaving animals (and will note so in the conclusions)  and would love to test and maybe improve our methods to datasets like this!
>
> 7. Section 4 organization: We will make the modifications if space allows.
>
> 8. Moving hyperparameters to the main text: Same as 7, we will make the modifications if space allows.
>
> 9. Video generation: we will add motif transitions in Figure 8 in the camera-ready version.

---

### Author Response · Authors · 2021-08-10
**General Response**

We thank all reviewers for their helpful comments and feedback on the paper! We have replied to individual reviewer comments below.

---

### Decision · Program_Chairs · 2021-09-28

**Decision:**

Accept (Spotlight)

**Comment:**

This paper proposes to learn disentagled embeddings of behavioural video---between time-dependent dynamic factors (called pose) and other factors (called context) in an unsupervised manner.

The motivation, model setup, objective, and experimental setup are all very well described, evaluated, and analysed. The model in particular is motivated very well given the specific application in consideration (behavioural video), and the ablations performed serve to highlight the advantages of the main components. The analyses also serve to highlight the effects of the disentanglement provided by the model.

I would strongly encourage the authors to carefully consider the reviewers' comments about edits and corrections, and ensure that these are incorporated in the updated manuscript.

Overall, the reviewers agree that this is a very good piece of work and based on these merits, the paper should be accepted for publication.


**Consistency Experiment:**

NeurIPS has a long history of experimentation. In 2014, NeurIPS ran an experiment in which 10% of submissions were reviewed by two independent committees to quantify the randomness in the review process. This year, we repeated a variant of this experiment to see how the quality of the review process has changed over time.  This paper was part of the experiment and was therefore assigned to two committees (consisting of reviewers, an Area Chair, and a Senior Area Chair) that reached independent decisions.  If both committees made the same recommendation, this recommendation was followed. If a single committee recommended acceptance, the paper was accepted (with the exception of a few cases in which the other committee identified what we considered a fatal flaw, e.g., an error in a key result).

This copy’s committee reached the following decision: **Accept (Spotlight)**

The other committee assigned to the paper recommended **Reject**.  You can find the other set of reviews, along with any follow up discussion with the authors here:
https://openreview.net/forum?id=I2pS-Lg7Xl